# EQUIVARIANT PROTEIN MULTI-TASK LEARNING

## ABSTRACT

Understanding and leveraging the 3D structures of proteins is central to various tasks in biology and drug discovery. While deep learning has been applied successfully for modeling protein structures, current methods usually employ distinct models for different tasks. Such a single-task strategy is not only resource-consuming when the number of tasks increases but also incapable of combining multi-source datasets for larger-scale model training, given that protein datasets are usually of small size for most structural tasks. In this paper, we propose to adopt one single model to address multiple tasks jointly, upon the input of 3D protein structures. In particular, we first construct a standard multi-task benchmark called PROMPT, consisting of 6 representative tasks integrated from 4 public datasets. The resulting benchmark contains partially labeled data for training and fully-labeled data for validation/testing. Then, we develop a novel graph neural network for multi-task learning, dubbed **He**terogeneous **M**ultichannel **E**quivariant **Net**work (HeMeNet), which is equivariant to 3D rotations/translations/reflections of proteins and able to capture various relationships between different atoms owing to the heterogeneous multichannel graph construction of proteins. Besides, HeMeNet is able to achieve task-specific learning via the task-aware readout mechanism. Extensive evaluations verify the effectiveness of multi-task learning on our benchmark, and our model generally surpasses state-of-the-art models. Our study is expected to open up a new venue for structure-based protein learning.

## 1 INTRODUCTION

Proteins are the workhorses of biological systems, performing a myriad of vital functions within cells and organisms. In recent years, learning-based methods have been applied widely to represent and leverage the 3D structures of proteins for various tasks including property prediction (Wang et al., 2023a), affinity prediction (Li et al., 2021), rigid docking (Ganea et al., 2022), and generation Kong et al. (2023), owing to their more efficient and reproducible implementation compared to those wet-lab experimental approaches. A major part of learning-based methods resort to Graph Neural Networks (GNNs) (Zhou et al., 2020; Xu et al., 2019), which naturally encode the 3D structures of proteins by modeling atoms or residues as nodes, and the connections in between as edges. In addition, certain GNNs are geometry-aware and they are E(3) equivariant/invariant in terms of arbitrary rotation/reflection/translation of the input structures, enabling the promising generalization ability across different choices of 3D coordinate systems (Satorras et al., 2021; Huang et al., 2022; Brandstetter et al., 2022).

In spite of the fruitful progress, existing methods usually employ one model for one task. A clear drawback of such a single-task training strategy is that it is resource-consuming, as the model should be re-trained for a newly coming task. More importantly, it is unable to combine multi-source datasets for model training. It is well known that structural datasets are often limited in size due to the expensive cost of collecting protein 3D structures and labels via wet lab experiments. For example, in PDBbind (Wang et al., 2004), a binding affinity database, there are only 5316 complexes for the Ligand Binding Affinity (LBA) refined set and 2852 complexes for the Protein-Protein Interaction (PPI) set. Considering that deep learning models nowadays are typically equipped with thousands of hundreds of parameters, conducting model training on a single-task dataset of small size usually leads to defective generalization ability.

In this paper, we propose protein multi-task learning: one model for multiple tasks upon 3D protein structures. We expect to increase the number of training samples by leveraging multi-task datasets and hence improve the generalizability and universality of the model. Nevertheless, accomplishing multi-task training is challenging. The first challenge is that there is even no available benchmark for multi-task training and evaluation. The ideal benchmark should cover a sufficient range of data and task types, and it should also contain a fully-labeled subset to better compare how a model performs on the same input for different task outputs. Therefore, constructing a benchmark of protein multi-task learning is far more difficult beyond a simple combination of current public datasets. The second challenge is that it is nontrivial to design a model that is generalist to serve our purpose. The model we desire should be capable of processing the complicated 3D structures of input proteins of various types, and it should also exhibit generally-promising performance across different tasks.

To achieve protein multi-task learning, we make the following contributions:

- To the best of our knowledge, we are the first to propose the concept of structure-based protein multi-task learning. To fulfil our purpose, we carefully integrate the structures and labels from 4 public datasets, and construct a new benchmark named **PRO**tein **M**ulti**P**le **T**ask (**PROMPT**), which consists of 6 representative tasks upon 3 different types of inputs, including single-chain proteins, protein-protein and ligand-protein complexes.

- We propose a novel model for protein structure learning, dubbed **He**terogeneous **M**ultichannel **E**quivariant Network (**HeMeNet**), which is equivariant to 3D rotations/translations/reflections of proteins and able to capture various relationships between different atoms owing to the heterogeneous multichannel graph construction of proteins. Additionally, HeMeNet develops a task-aware readout mechanism by associating the output head of each task with a learnable task prompt.

- For the experiments on PROMPT, our HeMeNet surpasses other state-of-the-art methods in most tasks under both the single-task and multi-task settings. Particularly on the LBA and PPI tasks, we find that the multi-task version of HeMeNet is significantly better than its single-task counterpart, verifying the benefit of multi-task learning in these cases.

## 2 RELATED WORKS

**Protein Interaction and Property Prediction**   Predicting the binding affinity and functional properties for proteins with computational methods is of growing interest (Wang et al., 2022; Zhao et al., 2020). Previous research learns protein representations by different forms of information, most of which take amino acid sequence (Alley et al., 2019; Rao et al., 2019), multiple sequence alignment (Rao et al., 2021) or 3D structure (Hermosilla et al., 2021; Zhang et al., 2023) as input. Many works encode the information of a protein's 3D structure by graph neural network (Gligorijević et al., 2021a; Zhang et al., 2023; Morehead et al., 2022). Li et al. (2021) take full-atom geometry at the interaction pocket/interface, and Zhang et al. (2023) take residue-level geometry of the whole protein for property prediction. Our method utilizes full-atom geometry on the whole protein to address interaction and property prediction tasks together.

**Equivariant Graph Neural Networks**   Many equivariant graph neural networks have emerged recently with the inductive bias of 3D symmetry, modeling various tasks including docking, molecular function prediction and sequence design (Thomas et al., 2018; Gasteiger et al., 2020; Satorras et al., 2021; Brandstetter et al., 2022). To empower the model with the ability to handle the complicated full-atom geometry, our model is closely related to dyMEAN (Kong et al., 2023), a multi-channel equivariant network that can deal with a dynamic number of channels in the protein graph. We propose a more powerful heterogeneous equivariant network that is capable of handling various incoming message types.

**Protein Multi-Task Learning**   Multi-task learning takes advantage of knowledge transfer from multiple tasks, achieving a better generalization performance. In the field of protein, several works leverage multi-task learning on the task of interaction prediction and property prediction, most of which are for sequence-based models. Lin et al. (2022) propose a pre-training method to learn protein and ligand representation, together with a multi-task dual adaptation mechanism for better LBA prediction from sequences. Shi et al. (2023) design three Enzyme Commission number related hierarchical tasks to train the model. Moon & Kim (2022) take a knowledge distillation manner to transfer knowledge from single-task models for DTA prediction. Wang et al. (2023b) introduce a

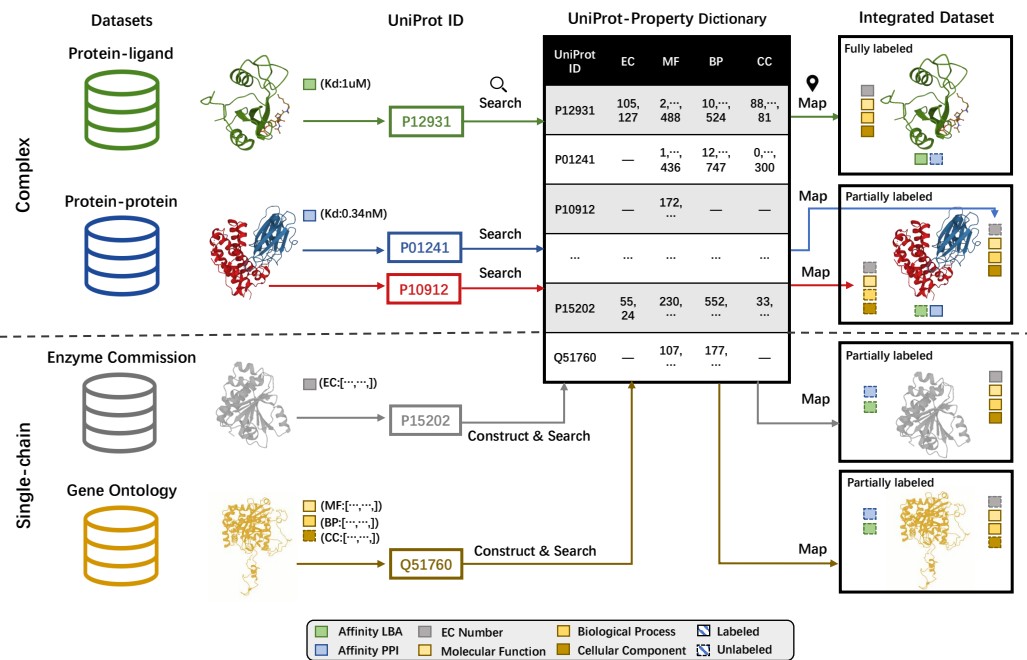

Figure 1: **Construction of PROMPT**. We first extract the UniProt ID for each chain by querying PDB API and construct a UniProt-Property dictionary to map the UniProt ID of each protein chain with EC and GO-MF, GO-BP, GO-CC labels annotated in the EC and GO datasets. With this dictionary, we can extract the UniProt ID for each chain, search it in the dictionary and map the chain with its labels. The complex with one affinity label and all property labels for each chain is defined as fully-labeled. We take most of the fully-labeled data for val/test, a few of the fully-labeled and most of the partially labeled data for training.

multi-task protein pre-training method with prompts. To the best of our knowledge, we are the first to combine structure-based interaction prediction and property prediction in a multi-task setting.

## 3 DATASET CONSTRUCTION: PROMPT

We construct a new dataset called PROtein MultiPle Tasks (PROMPT) for protein representation learning. PROMPT is composed of different types of tasks on 3D protein structures: the prediction of Ligand Binding Affinity (**LBA**) and Protein-Protein Interaction (**PPI**) based on two-instance complexes and the prediction of Enzyme Commission (**EC**) number and Gene Ontology (**GO**) terms based on single-chain structures. Particularly, the LBA and PPI tasks originated from the PDBbind database (Wang et al., 2004) aim at regressing the affinity value of a protein-ligand complex and protein-protein complex, respectively. The EC task is initially constructed by (Gligorijević et al., 2021b) to describe the catalysis of biochemical reactions, and it contains 19201 protein chains with 538 binary-class labels. The GO task consisting of 36641 annotated protein chains predicts the hierarchically-related functional properties of gene products (Gligorijević et al., 2021b): Molecular Function (MF), Biological Process (BP) and Cellular Component (CC). We treat the prediction of MF, BP and CC as three individual tasks, resulting in six different prediction tasks in total.

One key difficulty in integrating these tasks from their sourced datasets is that samples from one task may lack the labels for other tasks. It is crucial to obtain samples with a complete set of labels across tasks for the training and evaluation of multi-task learning methods. To address this, we propose a matching pipeline that enables us to transfer the labels between EC and GO, and assign EC and GO labels for the chains of complexes in LBA and PPI as well (it is impossible to conduct the inverse direction since it is meaningless to assign LBA or PPI affinity for those single chains in

EC and GO). Specifically, we utilize the UniProt ID to uniquely identify a protein chain[1]. We first obtain the UniProt IDs of all protein chains in PROMPT from Protein Data Bank (Berman et al., 2000). For each UniProt ID, we then determine the EC and GO properties based on the labels of the corresponding chains in the EC and GO datasets, resulting in a UniProt-Property dictionary. With this dictionary, for a chain missing EC or GO labels (e.g., a chain of a complex in LBA and PPI), we can supplement the missing labels by searching the UniProt-Property dictionary by its UniProt ID to retrieve any known EC and GO labels. We define a complex (from either LBA or PPI) as fully-labeled if the complex has one affinity label (LBA or PPI) and four function labels for each of its chains. After our above matching process, we yield 1327 fully-labeled complexes.

We formulate the train/validation/test split in terms of the chain-level sequence identity, through the alignment methods commonly used in single-chain property prediction tasks (Gligorijević et al., 2021b). Specifically, we employ MMSeq2 (Steinegger & Söding, 2017) to cluster all the chains in PROMPT with an alignment coverage $> 30\%$ and sequence identity of the aligned fragment $> 90\%$, leading to 33704 chain-level clusters. Then, we merge the clusters that contain the chains belonging to the same complex and finally get 30034 clusters. For the fully-sampled complexes, we randomly split them into the training, validation and test sets, with the number of complexes as 328, 530, and 469, respectively. For the partially labeled samples, we only retain those samples located in clusters different from the above test complexes and add them into the training set, resulting in an augmented training set with a total of 31252 samples. The statistics of the PROMPT dataset are listed in Table 5 and more details are presented in Appendix A.

## 4 METHOD

In this section, we first introduce our heterogeneous graph representation and the multi-task formulation in Section 4.1. Then, we design the architecture of the proposed HeMeNet in Section 4.2, which consists of two key components: heterogeneous multi-channel equivariant message passing and task-aware readout.

### 4.1 HETEROGENEOUS GRAPH REPRESENTATION AND TASK FORMULATION

The input of our model is of various types. It could be either a two-instance complex (protein-ligand for LBA and protein-protein for PPI) or a single chain (for EC and GO). Here, for consistency, we unify these two different kinds of input as a graph $\mathcal{G}$ composed of two sets of nodes $\mathcal{V}_r$ and $\mathcal{V}_l$. For the LBA complex input, $\mathcal{V}_r$ and $\mathcal{V}_l$ denote the receptor and the ligand, respectively, while for the PPI complex and single-chain input, $\mathcal{V}_r$ refers to the input protein and $\mathcal{V}_l$ becomes an empty set. Each node in $\mathcal{V}_r$ and $\mathcal{V}_l$ corresponds to a residue of the input protein or an atom of the input small molecular. We associate each node $v_i$ with the representation $(\boldsymbol{h}_i, \vec{\boldsymbol{X}}_i)$, where $\boldsymbol{h}_i \in \mathbb{R}^d$ denotes the node feature and it is initialized as a learnable residue embedding, $\vec{\boldsymbol{X}}_i \in \mathbb{R}^{3 \times c_i}$ indicates the 3D coordinates of all $c_i$ atoms within the node. As for edge construction, we borrow the ideas from GearNet (Zhang et al., 2023). In detail, for residue nodes, we allow $R$ heterogeneous types of edge connections including sequential edges of different distances ($d = \{-2, -1, 1, 2\}$), self-loop edges, and knn edges; for single-atom nodes, only knn edges are created. Note that different from GearNet, we expand the graph to the full-atom setting by involving all atom coordinates as input, rather than only using alpha-Carbon atoms. We present an example from the LBA task in Figure 2, where we only draw a few nodes and omit the self-loop edges except for the central node for simplicity. Overall, we obtain a full-atom heterogeneous graph representation $\mathcal{G}$ for each input.

**Task Formulation** Given a full-atom heterogeneous graph $\mathcal{G}$, our goal is to design a model $\boldsymbol{p} = f(\mathcal{G})$ with multiple-dimensional output $\boldsymbol{p}$ that is able to predict the complex-level affinity and chain-level functional properties simultaneously. By making use of our proposed dataset PROMPT, we train the model with a partially labeled training set (containing three possible kinds of input: protein-protein graph, protein-ligand graph and single-chain graph) and test it on the fully-labeled test set (only containing protein-protein graph or protein-ligand graph). Notably, the prediction should be invariant with regard to any E(3) transformation (rotation/reflection/translation) of the in-

---

[1]The UniProt dataset is the world's leading non-redundant protein sequence and function dataset and it identifies proteins by their UniProt IDs.

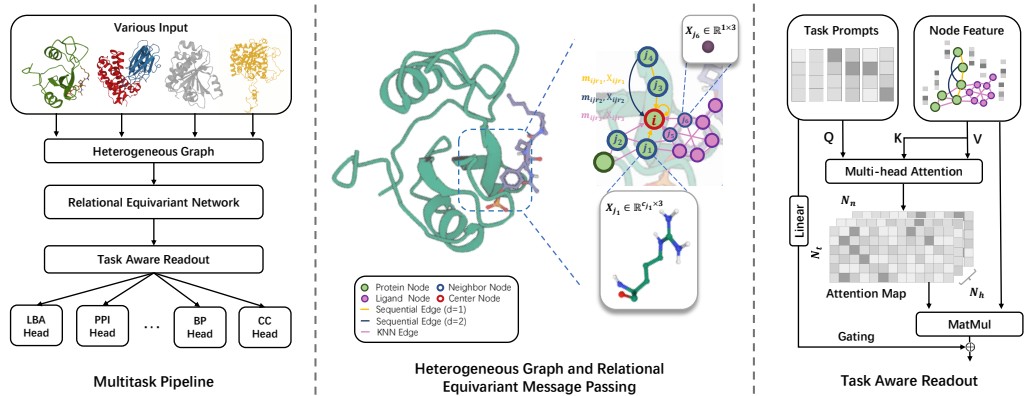

Figure 2: **Overview of our pipeline**. Left: The pipeline of our multi-task learning network. HeMeNet takes two-instance complexes or a single chain as input and predicts complex-level affinity and chain-level properties simultaneously. Middle: An example of the heterogeneous graph representation and the relational equivariant message passing. We only annotate a small part of our multi-channel full-atom graph for simplicity. Each edge is bidirectional, and we only mark the incoming edge arrow and self-loop for the center node. Right: Task-aware readout module. We take a task prompt as the query for each task, generating attention maps for all the nodes to get a suitable readout for different downstream tasks.

put coordinates. To do so, we will formulate equivariant layers plus an invariant output layer in our model, which will be detailed in the next subsection.

## 4.2 HeMeNet: Heterogeneous Multi-channel Equivariant Network

To better cope with the 3D structures of different types for different tasks, we propose a heterogeneous multi-channel E(3) equivariant graph neural network with the ability to aggregate different relational messages. After several layers of the message passing, the node representations are transformed into task-specific representations by a task-aware readout module, generating appropriate complex-level and chain-level predictions via different task heads.

**Heterogeneous Multi-channel Equivariant Message Passing** Similar to dyMEAN (Kong et al., 2023), we leverage a multi-channel coordinate matrix with varying size to record the geometric information of a node in an input graph. Moreover, in contrast to dyMEAN, we conduct heterogeneous message passing along multiple types of edges to capture rich relationships between nodes. We denote the node feature and coordinates as $(\boldsymbol{h}_i^{(l)}, \vec{\boldsymbol{X}}_i^{(l)})$ in the $l$-th layer. The message passing is calculated as:

$$\boldsymbol{m}_{ijr} = \phi_m(\boldsymbol{h}_i^{(l)}, \boldsymbol{h}_j^{(l)}, \frac{T_R(\vec{\boldsymbol{X}}_i^{(l)}, \vec{\boldsymbol{X}}_j^{(l)})}{||T_R(\vec{\boldsymbol{X}}_i^{(l)}, \vec{\boldsymbol{X}}_j^{(l)})||_F + \epsilon}, \boldsymbol{e}_r), \tag{1}$$

$$\vec{\boldsymbol{M}}_{ijr} = T_S(\vec{\boldsymbol{X}}_i^{(l)} - \frac{1}{c_j}\sum_{k=1}^{c_j}\vec{\boldsymbol{X}}_j^{(l)}(:,k), \phi_x(\boldsymbol{m}_{ijr})), \tag{2}$$

where, $\boldsymbol{m}_{ijr}$ and $\vec{\boldsymbol{M}}_{ijr}$ are separately the invariant and equivariant messages from node $j$ to $i$ along the $r$-th edge; $\boldsymbol{e}_r$ is the edge embedding feature; $\phi_m, \phi_x$ are Multi-Layer Perceptrons (MLPs) (Gardner & Dorling, 1998) with one hidden layer; $|| \cdot ||_F$ computes the Frobenius norm, $T_R$ and $T_S$ are the adaptive multichannel geometric relation extractor and geometric message scaler proposed in dyMEAN (Kong et al., 2023), in order to deal with the issue incurred by the varying shape of $\vec{\boldsymbol{X}}_i^{(l)}$ and $\vec{\boldsymbol{X}}_j^{(l)}$ since the number of atoms could be different for different nodes. With the calculated

messages, the node representation is updated by:

$$\boldsymbol{h}_i^{(l+1)} = \boldsymbol{h}_i^{(l)} + \sigma(\text{BN}(\phi_h(\sum_{r \in R} \boldsymbol{W}_r \sum_{j \in \mathcal{N}_r(i)} \boldsymbol{m}_{ijr}))), \tag{3}$$

$$\vec{\boldsymbol{X}}_i^{(l+1)} = \vec{\boldsymbol{X}}_i^{(l)} + \frac{1}{\sum_{r \in R} |\mathcal{N}_r(i)|} \sum_{r \in R} \sum_{j \in \mathcal{N}_r(i)} w_r \vec{\boldsymbol{M}}_{ijr}, \tag{4}$$

where, $\boldsymbol{W}_r, w_r$ are a learnable matrix and a learnable scalar, respectively, for the $r$-th kind of edge; $\mathcal{N}_r(i)$ denotes the neighbor nodes of $i$ regarding the $r$-th kind of edges; $\phi_h$ is an MLP, BN is the batch normalization operation, and $\sigma$ is an activation function. During the message-passing process, our model gathers information from different relations for $\boldsymbol{h}_i$ and $\vec{\boldsymbol{X}}_i$, ensuring the E(3) equivariance. For further details of the components in our model, please refer to Appendix B

**Task-Aware Readout**    After $L$ layers of the above relational message passing, we attain a set of E(3) invariant node features $\boldsymbol{H}^{(L)} \in \mathbb{R}^{n \times d_L}$, where $n$ is the number of nodes and $d_L$ is the feature dimension. To correlate the task-specific information with each node feature, we propose a task-aware readout function. We compute the attention weights between each node and each task-specific query, and then readout all weighted node features into a graph-level or chain-level representation for each task. As shown in Figure 2, the task-aware readout module is formulated as

$$\boldsymbol{\alpha}_t = \text{Softmax}(\frac{\boldsymbol{K}\boldsymbol{q}_t}{\sqrt{d_L}}), \tag{5}$$

$$\boldsymbol{f}_t = \text{FFN}(\boldsymbol{\alpha}_t \boldsymbol{V} + \text{Linear}(\boldsymbol{q}_t)), \tag{6}$$

where, $\boldsymbol{\alpha}_t \in [0, 1]^n$ defines the attention values for task $t$; $\boldsymbol{q}_t \in \mathbb{R}^{d_L}$ is the learnable query for task $t$; $\boldsymbol{K} = \boldsymbol{H}\boldsymbol{W}_K \in \mathbb{R}^{n \times d_L}$ and $\boldsymbol{V} = \boldsymbol{H}\boldsymbol{W}_V \in \mathbb{R}^{n \times d_L}$ are the key and value matrices, respectively; FFN is the feed-forward network containing layer normalization and linear layers; $\text{Linear}(\boldsymbol{q}_t) = \boldsymbol{W}_Q\boldsymbol{q}_t + \boldsymbol{b}$ is used before the shortcut addition. In our implementation, we apply the multi-head attention strategy by defining multiple query vectors for each task. Although we compute the attention by only using the invariant features $\boldsymbol{H}^{(L)}$, it indeed has involved the geometric information from the 3D coordinates during the previous $L$-layer message passing.

**Multiple Task Heads**    We feed the above task-specific feature $\boldsymbol{f}_i$ into different task heads implemented by MLPs, resulting in a prediction list $(\boldsymbol{p}_{\text{lba}}, \boldsymbol{p}_{\text{ppi}}, \boldsymbol{p}_{\text{ec}}, \boldsymbol{p}_{\text{mf}}, \boldsymbol{p}_{\text{bp}}, \boldsymbol{p}_{\text{cc}})$. For regression tasks (including LBA and PPI), we use the Mean Square Error (MSE) loss $\mathcal{L}_{\text{MSE}}$. For classification tasks (including EC, GO-MF, GO-BP and GO-CC), we use the Binary Cross Entropy (BCE) loss $\mathcal{L}_{\text{BCE}}$. The training loss is formulated as:

$$\mathcal{L} = \mathcal{L}_{\text{reg}} + \lambda\mathcal{L}_{\text{cls}}, \tag{7}$$

where $\mathcal{L}_{\text{reg}} = \mathbf{1}_{\text{lba}}\mathcal{L}_{\text{MSE}}(\boldsymbol{p}_{\text{lba}}) + \mathbf{1}_{\text{ppi}}\mathcal{L}_{\text{MSE}}(\boldsymbol{p}_{\text{ppi}}), \mathcal{L}_{\text{cls}} = \mathbf{1}_{\text{ec}}\mathcal{L}_{\text{BCE}}(\boldsymbol{p}_{\text{ec}}) + \mathbf{1}_{\text{mf}}\mathcal{L}_{\text{BCE}}(\boldsymbol{p}_{\text{mf}}) + \mathbf{1}_{\text{bp}}\mathcal{L}_{\text{BCE}}(\boldsymbol{p}_{\text{bp}}) + \mathbf{1}_{\text{cc}}\mathcal{L}_{\text{BCE}}(\boldsymbol{p}_{\text{cc}}), \lambda$ is a hyper-parameter to balance the trade-off of the losses. To allow training on the partially labeled sample, if the label of the task $*$ exists, then $\mathbf{1}_* = 1$, otherwise $\mathbf{1}_* = 0$. In addition, we adopt a balanced sampling strategy to ensure that each sampled mini-batch should contain at least one sample labeled for each task, which is shown to further accelerate the training convergence in our experiments.

## 5 EXPERIMENTS

In this section, we will first introduce the experimental setup in Section 5.1. In Section 5.2, we evaluate our model on the proposed challenging dataset PROMPT for affinity and property prediction in both single-task and multi-task settings and compare it with other baseline models. In Section 5.3, we experiment with different readout strategies and compare their performance on property prediction tasks. In Section 5.4, we perform ablation experiments on different modules.

### 5.1 EXPERIMENTAL SETUP

**Task settings**    We compare the performance of HeMeNet with other models under single-task and multi-task settings using the same validation and test sets. For single-task training, models are

trained on samples with labels of the corresponding task. We also remove the task-aware readout of our model for a fair comparison. For multi-task training, the models are trained on all partially labeled training samples.

During distributed training, for each batch on each node, we use a balanced sampler to sample at least one complex from LBA and one complex from PPI. We include samples with up to 15,000 atoms for training and evaluation. All models are trained for 30 epochs on 4 NVIDIA A100 GPUs with a batch size of 8 for each GPU. More detail on experimental settings can be found in Appendix C

**Baselines** We compared our model with seven representative baselines. **GCN** (Kipf & Welling, 2017) aggregates information weighted by the degree of nodes. **GAT** (Veličković et al., 2018) utilizes an attention mechanism for message passing. **Schnet** (Schütt et al., 2017) is an invariant network with continuous filter convolution on the 3D molecular graph. **GearNet** (Zhang et al., 2023) designs a relational message-passing network to capture information on protein function tasks. Besides the previous invariant models, we also compare our method with equivariant models. **EGNN** (Satorras et al., 2021) is a lightweight but effective E(n) equivariant graph neural network. **GVP** (Jing et al., 2021) designs an equivariant geometric vector perceptron for protein representation. **dyMEAN** (Kong et al., 2023) is an equivariant model for antibody design; it takes a dynamic multichannel equivariant function for full-atom coordinates. For GearNet, we take alpha-Carbon atoms as input following the original paper, and we take full-atom graphs for other models.

**Evaluation** For LBA and PPI tasks, we employ the commonly used Root Mean Square Error (RMSE) and Mean Average Error (MAE) as the evaluation metrics (Townshend et al., 2021). For EC and GO tasks, we use maximum F-score (Fmax) following (Zhang et al., 2023). Each experiment is independently run three times with different random seeds.

## 5.2 RESULTS ON PROMPT

We conduct experiments under both single-task and multi-task settings. The mean results of three runs are reported in Table 1. Results with standard deviation can be found in Appendix D. According to the results, we draw conclusions summarized in the subsequent paragraphs.

**Our model outperforms the baselines on most of the tasks under both settings.** Under the single-task setting, our model surpasses other models in five of the six tasks. Under the multi-task setting, our model surpasses other models in four of the six tasks, with the remaining two tasks reaching second and third place, respectively. These results demonstrate the effectiveness of our heterogeneous graph construction and relational equivariant message passing for handling the full-atom input. Notably, under the single-task setting, only the models with a heterogeneous message passing (GearNet and ours) can perform well on all of the four property prediction tasks. For the single-task PPI prediction, our model and other full-atom models are struggling due to the small size of the PPI training set and the noise introduced by processing the atoms of the whole protein. Under the multi-task setting, our full-atom model, benefiting from joint learning, shows superior results on different tasks, and there are two main interesting observations discussed next.

**Our model benefits from the multi-task setting, especially on LBA and PPI.** We observe that almost all models improve their performance on LBA and PPI tasks under the multi-task setting. In particular, our model significantly improves the PPI RMSE from 6.031 to 1.087 by utilizing a training set that is more than ten times larger (2587 for PPI single-task and 30904 for our multi-task training set). We also train our model with alpha C atom (Hemenet w/$C_\alpha$) as input, resulting in a best PPI RMSE of 0.861. The performance on LBA is also improved by 0.182 for RMSE. Moreover, multi-task learning also reduces the standard deviation of the regression error (see Table 6). These results demonstrate that the model can handle challenging tasks (complex-level affinity prediction) better when more structural information is available (e.g., single-chain structures and their labels).

**Our model performs harmonious multi-task training on property prediction tasks.** We observe that when switching from the single-task to multi-task setting, baseline models experience performance degradation to some extent across the four property prediction tasks. This is a common phenomenon in multi-task learning due to task interference among diverse tasks, and combining different tasks for training without careful adaption can harm performance. With the guidance of our task-aware readout module, our model is able to learn from multiple tasks in a task-harmonious

Table 1: The mean result for three runs on the full-label test set. We select representative invariant and equivariant models for affinity prediction and property prediction. The upper half reports the results for the single-task setting, and the lower half reports the results for the multi-task setting. The best results are marked in bold and the second best results are underlined. In the multi-task setting, we train the models with the same size compared to their corresponding single-task models.

| | Method | LBA | | PPI | | EC↑ | GO | | |
|---|---|---|---|---|---|---|---|---|---|
| | | RMSE↓ | MAE↓ | RMSE↓ | MAE↓ | | MF↑ | BP↑ | CC↑ |
| Single-task | GCN (Kipf & Welling, 2017) | 2.193 | 1.721 | 7.840 | 7.738 | 0.022 | 0.207 | 0.254 | 0.367 |
| | GAT (Veličković et al., 2018) | 2.301 | 1.838 | 7.820 | 7.720 | 0.018 | 0.223 | 0.249 | 0.354 |
| | SchNet (Schütt et al., 2017) | 2.162 | 1.692 | 7.839 | 7.729 | 0.097 | 0.311 | 0.281 | 0.431 |
| | GearNet (Zhang et al., 2023) | 1.957 | 1.542 | **2.004*** | **1.279*** | 0.716 | 0.677 | 0.252 | 0.438 |
| | EGNN (Satorras et al., 2021) | 2.282 | 1.849 | 4.854 | 4.756 | 0.039 | 0.206 | 0.253 | 0.357 |
| | GVP (Jing et al., 2021) | 2.281 | 1.789 | 5.280 | 5.267 | 0.020 | 0.204 | 0.244 | 0.454 |
| | dyMEAN (Kong et al., 2023) | 2.410 | 1.987 | 7.309 | 7.182 | 0.115 | 0.436 | 0.292 | 0.477 |
| | Hemenet (Ours) | **1.912** | **1.490** | 6.031 | 5.891 | **0.863** | **0.778** | **0.404** | **0.544** |
| Multi-task | SchNet (Schütt et al., 2017) | 1.763 | 1.447 | 1.216 | 1.120 | 0.093 | 0.192 | 0.264 | 0.402 |
| | GearNet (Zhang et al., 2023) | 2.193 | 1.863 | 1.275 | 1.035 | 0.187 | 0.203 | 0.261 | 0.379 |
| | EGNN (Satorras et al., 2021) | 1.777 | 1.441 | 0.999 | 0.821 | 0.048 | 0.169 | 0.244 | 0.352 |
| | GVP (Jing et al., 2021) | 1.870 | 1.572 | 0.906 | 0.758 | 0.018 | 0.168 | 0.246 | 0.360 |
| | dyMEAN (Kong et al., 2023) | 1.777 | 1.446 | 1.725 | 1.523 | 0.038 | 0.164 | 0.263 | **0.449** |
| | Hemenet w/$C_\alpha$ (Ours) | 1.799 | 1.420 | **0.861** | **0.719** | 0.630 | 0.595 | 0.279 | 0.426 |
| | Hemenet (Ours) | **1.730** | **1.335** | 1.087 | 0.912 | **0.810** | **0.727** | **0.379** | 0.436 |

* In the single-task setting, Only GearNet and is trained under the alpha-Carbon atom only setting and it significantly outperforms other baselines on PPI. We suppose this is because the training set for PPI is quite small, and there is too much noise when considering the full atoms of the whole protein. The alpha-only input simplifies single-task PPI learning.

way, while achieving performance on the property prediction tasks comparable to their single-task counterparts with the same parameter size.

## 5.3 COMPARISON OF DIFFERENT READOUT METHODS

To verify the effectiveness of our task-aware readout module, we take HeMeNet and Gearnet as the backbone and compare the task-aware readout method with two commonly used readout functions: sum readout and weight the node feature by multiplying them with a task prompt (Liu et al., 2023). The results are presented in Table 2. We can conclude the following observations: 1) Our proposed task-aware readout model effectively injects task-related information using an attention mechanism, leading to overall improvements for various tasks, especially on the Enzyme Commission task. 2) Simply element-wise multiplication of the same task prompt feature with all the nodes fails to provide sufficient guidance to improve performance across all tasks.

To further investigate the relationship between multiple task prompts, we calculate Pearson's correlation between prompts. As shown in Figure 3, the correlations between tasks within the same category (e.g. EC and MF) are high, while the correlations between tasks from different categories (e.g. LBA and BP) are low. A high correlation between prompts indicates similar attention queries, leading to similar readout functions. This suggests that with the task-aware guidance, the model employs similar readout strategies for tasks from the same category and divergent strategies from tasks from different categories.

## 5.4 ABLATION STUDY

We perform ablation experiments to evaluate the necessity of different components, including the task-aware readout, the relational message passing mechanismand the full-atom geometry. Specifi-

| Method | EC↑ | GO-MF↑ | GO-BP↑ | GO-CC↑ |
|---|---|---|---|---|
| **Gearnet**$_s$ | 0.187 | 0.203 | 0.261 | 0.379 |
| **Gearnet**$_w$ | 0.066 | 0.164 | 0.271 | 0.414 |
| **Gearnet**$_t$ | 0.421 | 0.310 | 0.287 | 0.403 |
| **Hemenet**$_s$ | 0.722 | 0.558 | 0.302 | 0.413 |
| **Hemenet**$_w$ | 0.325 | 0.312 | 0.276 | **0.440** |
| **Hemenet**$_t$ | **0.810** | **0.727** | **0.379** | 0.436 |

Table 2: Comparison of different readout functions for multi-task learning. $s$, $w$, and $t$ represent sum, weighted feature and task-aware readout, respectively.

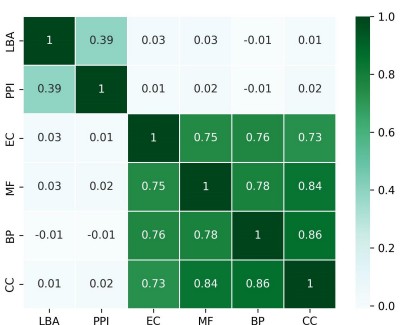

Figure 3: Prompt correlation

cally, the ablation of TAR replaces the task-aware readout module with a sum readout. For $e_r, W_r$ and $w_r$, we remove different types of edges and the corresponding relational message passing weights. For full-atom ablation, we represent the coordinates of residues by their alpha C atoms.

Table 3: Ablation study for different components in HeMeNet.

| Method | LBA | | PPI | | EC↑ | GO | | |
|---|---|---|---|---|---|---|---|---|
| | RMSE↓ | MAE↓ | RMSE↓ | MAE↓ | | MF↑ | BP↑ | CC↑ |
| HeMeNet | **1.730** | **1.335** | 1.087 | 0.912 | **0.810** | **0.727** | **0.379** | **0.436** |
| - TAR | 1.905 | 1.462 | 1.970 | 1.583 | 0.722 | 0.558 | 0.302 | 0.413 |
| - $e_r, W_r, w_r$ | 1.790 | 1.415 | 1.446 | 1.277 | 0.547 | 0.663 | 0.359 | 0.391 |
| - full-atom | 1.799 | 1.420 | **0.861** | **0.719** | 0.630 | 0.595 | 0.279 | 0.426 |

We present the results for ablation studies in Table 3, the observations are as follows: 1) Without our task-aware readout strategy, significant performance degradation are observed in all tasks, indicating that the tasks can hinder each other without appropriate guidance. 2) Without the heterogeneous graph representation and the corresponding relational message passing, our model's performance drops on property prediction tasks, especially the Enzyme Commission number prediction. 3) Removing the full-atom geometry decreases the performance in multiple tasks. However, it improves our model's performance in PPI. Similar to the explanation in Section 5.2, we suppose that the large number of atoms in the full-atom protein-protein complex introduces excessive noise to prediction compared with input with alpha-Carbon atoms only.

# 6 CONCLUSION

**Limitations** It is hard to add more tasks into PROMPT while maintaining the fully-labeled sample size under our definition of fully-labeled data. Relaxing the restriction of the test set can alleviate this issue. Meanwhile, the input is now a mixture of single chains and complexes, we can randomly augment chains from their original PDB complex to form 'complexes' and label them based on their UniProt IDs. Besides, we only consider invariant tasks in this work, we can also extend our model to more tasks in future work (e.g. equivariant tasks).

In this paper, we alleviate the problem of sparse data in structured protein datasets by a multi-task setting. We construct a standard multi-task benchmark PROMPT, consisting of 6 representative tasks integrated from 4 public datasets for joint learning. We propose a novel network called HeMeNet to address multiple tasks in protein 3D learning, with a novel heterogeneous equivariant full-atom encoder and a task-aware readout module. Comprehensive experiments demonstrate our model's performance on the affinity and property prediction tasks. Our work brings insights for utilizing different structural datasets to train a more powerful generalist model in future research.

## REPRODUCIBILITY

We provide detailed information about our dataset PROMPT in Appendix A, and we will make our dataset PROMPT and code public.

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

# A DATA

## A.1 DATASET SOURCES

### A.1.1 ENZYME COMMISSION AND GENE ONTOLOGY

We adopt the data set from (Gligorijević et al., 2021b). This data set contains 19201 PDB chains from 538 EC numbers, selected from the third and fourth levels of EC tree. The GO terms with at least 50 and not more than 5,000 training examples are selected. The number of classes in each branch of GO (MF, BP, CC) is 490, 1944 and 321, respectively. This data set consolidates 36641 PDB chains with their GO terms label. We obtain the structure of PDB chains from TorchDrug (Zhu et al., 2022).

### A.1.2 LBA AND PPI

We adopt the data set from PDBbind (Wang et al., 2004) (version 2020). This dataset contains 5316 protein-ligand complexes in the refined set and 2852 protein-protein complexes with known binding data (in the form of $K_d$, $K_i$, $IC_{50}$ values) manually collected from the original references (Wang et al., 2004). We obtain the structure of the complexes from PDBbind. The PDBbind dataset can be downloaded from `http://www.pdbbind.org.cn`.

## A.2 PROMPT STATSTICS

As described in section 3, after the procedure of labeling and splitting, we get a new dataset named PROMPT. Table 5 shows the detailed statistics of PROMPT in different tasks. Note that the sample number for multi-task is slightly different from that in Table 4 since we removed samples with atom numbers greater than 15,000.

Table 4: Dataset split. The fully-labeled data are randomly divided into the train, validation and test sets. Partially labeled samples located in the clusters different from the above test complexes are retained and added to the training set.

| Clusters after merged | Train set size | Validation set size | Test set size |
|---|---|---|---|
| 30034 | 31252 | 530 | 469 |

Table 5: Dataset details for different tasks. We summarize the number of samples that contains a specific task's annotation in PROMPT.

| Task Name | # Train samples | # Validation samples | # Test samples |
|---|---|---|---|
| multi-task | 30904 | 516 | 467 |
| LBA | 3247 | 493 | 452 |
| PPI | 1976 | 23 | 15 |
| EC | 15025 | 516 | 467 |
| GO-MF | 22997 | 516 | 467 |
| GO-BP | 21626 | 516 | 467 |
| GO-CC | 10543 | 516 | 467 |

# B  DETAILS FOR HeMeNet COMPONENTS

In this section, we mainly introduce $T_R$ and $T_S$ borrowed from (Kong et al., 2023) and our modification for the heterogeneous graph.

**The Geometric Relation Extractor** $T_R$ can deal with coordinate sets with different channels. Given $\boldsymbol{X}_i \in \mathbb{R}^{3 \times c_i}$ and $\boldsymbol{X}_j \in \mathbb{R}^{3 \times c_j}$, we can compute the channel-wise distance for each coordinate pairs: $D_{ij}(p, q) = ||\boldsymbol{X}_i(:, p) - \boldsymbol{X}_j(:, q)||_2$. Different from (Kong et al., 2023), we use two fixed binary vectors $\boldsymbol{w}_i \in \mathbb{R}^{c_i \times 1}$ and $\boldsymbol{w}_j \in \mathbb{R}^{c_j \times 1}$, when there is an element in the channel, its weight is set to 1, otherwise 0. We also adjusted the learnable attribute matrices $\boldsymbol{A}_i \in \mathbb{R}^{c_i \times d}$ and $\boldsymbol{A}_j \in \mathbb{R}^{c_j \times d}$ to be suitable to our input, assigning different element embedding for each channel. The final output $\boldsymbol{R}_{ij} \in \mathbb{R}^{d \times d}$ is given by:

$$\boldsymbol{R}_{ij} = \boldsymbol{A}_i^T (\boldsymbol{w}_i \boldsymbol{w}_j^T \odot \boldsymbol{D}_{ij}) \boldsymbol{A}_j. \tag{8}$$

$\boldsymbol{R}_{ij}$ keeps its shape awareness of $c_i$ and $c_j$.

**The Geometric Message Scaler** $T_S$ aims to generate geometric information of vary coordinate set $\boldsymbol{X} \in \mathbb{R}^{3 \times c}$ with the fixed length incoming message $s = \phi_x(\boldsymbol{m}_{ij}) \in \mathbb{R}^C$, where $C = 14$ is the max channel size of the common amino acids. Then, $T_S(\boldsymbol{X}, \boldsymbol{s})$ is calculated by:

$$\boldsymbol{X}' = \boldsymbol{X} \cdot \mathrm{diag}(s'), \tag{9}$$

where $s' \in \mathbb{R}^c$ is the average pooling of $\boldsymbol{s}$ with a sliding window of size $C - c + 1$ and stride 1, and $\cdot$ is a diagonal matrix with the input vector $s$ as the diagonal elements.

# C  IMPLEMENTATION DETAILS AND HYPERPARAMETERS

In this section, we introduce the implementation details of all baselines and our model. For all models, we concatenate the hidden output for the final output. For the multi-task setting, all the models except HeMeNet take the sum readout method. The feature after the readout function will be fed into six two-layer MLPs to make predictions for different prediction tasks. The input for models are full-atom with KNN edges, except for GearNet and HeMeNet.

**GCN (Kipf & Welling, 2017)**, **GAT (Veličković et al., 2018)** and **SchNet (Schütt et al., 2017)**. We take the implementation in PyTorch Geometric (Fey & Lenssen, 2019), with a 3-layer design. For all the models, the hidden size is set to 256.

**GearNet (Zhang et al., 2023)**. We re-implement GearNet with reference to its official implementation, with a six-layer design. The hidden size is set to 512, and the cutoff value is 4.5 following the original settings. For the multi-task setting, we take the sum readout method. We use the alpha-Carbon atom only graph for GearNet as the input.

**EGNN (Satorras et al., 2021)**. We re-implement EGNN with reference to its official implementation, with a 3-layer design. The hidden size is set to 256.

**GVP (Jing et al., 2021)**. We take the implementation in PyTorch Geometric (Fey & Lenssen, 2019), with a 3-layer design. The hidden size is set to 128 following the original implementation.

**dyMEAN (Kong et al., 2023)**. We re-implement dyMEAN with reference to its official implementation, with a 6-layer design. The hidden size is set to 256.

**HeMeNet (ours)**. We take a 6-layer design for our model, and the hidden size is set to 256. We take our task-aware readout module to generate features for different tasks. We use the full-atom heterogeneous graph for HeMeNet as the input.

Table 6: The detailed results for three runs on the full-label test set. We select representative invariant and equivariant models for affinity prediction and property prediction. The upper half reports the results for the single-task setting, and the lower half reports the results for the multi-task setting. The best results are marked in bold and the second best results are underlined. In the multi-task setting, we train the models with the same size compared to their corresponding single-task models.

| | Method | LBA | | PPI | | EC | GO | | |
|---|---|---|---|---|---|---|---|---|---|
| | | RMSE↓ | MAE↓ | RMSE↓ | MAE↓ | EC↑ | MF↑ | BP↑ | CC↑ |
| **Single-task** | GCN (Kipf & Welling, 2017) | 2.193 ± 0.004 | 1.721 ± 0.005 | 7.840 ± 0.044 | 7.738 ± 0.044 | 0.022 ± 0.017 | 0.207 ± 0.002 | 0.254 ± 0.009 | 0.367 ± 0.012 |
| | GAT (Veličković et al., 2018) | 2.301 ± 0.004 | 1.838 ± 0.013 | 7.820 ± 0.116 | 7.720 ± 0.112 | 0.018 ± 0.001 | 0.223 ± 0.009 | 0.249 ± 0.033 | 0.354 ± 0.002 |
| | SchNet (Schütt et al., 2017) | 2.162 ± 0.016 | 1.692 ± 0.030 | 7.839 ± 0.057 | 7.729 ± 0.056 | 0.097 ± 0.012 | 0.311 ± 0.021 | 0.281 ± 0.006 | 0.431 ± 0.026 |
| | Gearnet (Zhang et al., 2023) | 1.957 ± 0.044 | 1.542 ± 0.055 | **2.004 ± 0.213** | **1.279 ± 0.290** | 0.716 ± 0.026 | 0.677 ± 0.026 | 0.252 ± 0.029 | 0.438 ± 0.091 |
| | EGNN (Satorras et al., 2021) | 2.282 ± 0.082 | 1.849 ± 0.083 | 4.854 ± 0.679 | 4.756 ± 0.672 | 0.039 ± 0.025 | 0.206 ± 0.011 | 0.253 ± 0.009 | 0.357 ± 0.006 |
| | GVP (Jing et al., 2021) | 2.281 ± 0.005 | 1.789 ± 0.007 | 5.280 ± 0.041 | 5.267 ± 0.040 | 0.020 ± 0.006 | 0.204 ± 0.010 | 0.244 ± 0.008 | 0.454 ± 0.102 |
| | dyMEAN (Kong et al., 2023) | 2.410 ± 0.184 | 1.987 ± 0.181 | 7.309 ± 0.540 | 7.182 ± 0.543 | 0.115 ± 0.034 | 0.436 ± 0.506 | 0.292 ± 0.016 | 0.477 ± 0.056 |
| | Hemenet (Ours) | **1.912 ± 0.420** | **1.490 ± 0.050** | 6.031 ± 0.380 | 5.891 ± 0.380 | **0.863 ± 0.016** | **0.778 ± 0.003** | **0.404 ± 0.014** | **0.544 ± 0.024** |
| **Multi-task** | SchNet (Schütt et al., 2017) | 1.763 ± 0.019 | 1.447 ± 0.015 | 1.216 ± 0.033 | 1.120 ± 0.037 | 0.093 ± 0.008 | 0.192 ± 0.013 | 0.264 ± 0.004 | 0.402 ± 0.024 |
| | Gearnet (Zhang et al., 2023) | 2.193 ± 0.373 | 1.863 ± 0.364 | 1.275 ± 0.471 | 1.035 ± 0.197 | 0.187 ± 0.128 | 0.203 ± 0.107 | 0.261 ± 0.020 | 0.379 ± 0.031 |
| | EGNN (Satorras et al., 2021) | 1.777 ± 0.029 | 1.441 ± 0.046 | 0.999 ± 0.054 | 0.821 ± 0.057 | 0.048 ± 0.020 | 0.169 ± 0.009 | 0.244 ± 0.006 | 0.352 ± 0.010 |
| | GVP (Jing et al., 2021) | 1.870 ± 0.309 | 1.572 ± 0.034 | **0.906 ± 0.329** | **0.758 ± 0.015** | 0.018 ± 0.001 | 0.168 ± 0.003 | 0.246 ± 0.007 | 0.360 ± 0.016 |
| | dyMEAN (Kong et al., 2023) | 1.777 ± 0.075 | 1.446 ± 0.103 | 1.725 ± 0.362 | 1.523 ± 0.383 | 0.038 ± 0.014 | 0.164 ± 0.014 | 0.263 ± 0.011 | **0.449 ± 0.011** |
| | Hemenet w/$C_\alpha$ (Ours) | 1.799 ± 0.019 | 1.420 ± 0.040 | **0.861 ± 0.051** | **0.719 ± 0.068** | 0.630 ± 0.043 | 0.595 ± 0.037 | 0.279 ± 0.025 | 0.426 ± 0.042 |
| | Hemenet (Ours) | **1.730 ± 0.030** | **1.335 ± 0.031** | 1.087 ± 0.224 | 0.912 ± 0.204 | **0.810 ± 0.083** | **0.727 ± 0.084** | **0.379 ± 0.043** | 0.436 ± 0.053 |

# D  MORE INFORMATION ON THE EXPERIMENT RESULT

We show the standard deviation of three independent repeated results on PROMPT in Table 6. As represented in the table, the standard deviation of LBA and PPI tasks in the multi-task setting decreases significantly than in the single-task setting.

