# OpenReview forum: "Equivariant Protein Multi-task Learning"
_ICLR.cc/2024/Conference — Submitted to ICLR 2024_

### Official Review · Reviewer_N1C4 · 2023-10-24

**Soundness:** 3 good
**Presentation:** 2 fair
**Contribution:** 2 fair
**Rating:** 5
**Confidence:** 4

**Summary:**

This work proposes a multi-task learning framework for structure-based protein function prediction. At the first place, authors contribute a multi-task benchmark dataset PROMPT that incorporates protein-ligand affinity prediction, protein-protein affinity prediction and protein function prediction. For addressing the multi-task learning under such a set, authors employ a multi-relational equivariant encoder for extracting protein representations and use task-specific heads for prediction. Based the experimental results, the proposed architecture outperforms previous architectures (like GVP and GearNet) on single-task learning, and some benefits are observed on ligand-protein and protein-protein affinity prediction by applying multi-task learning.

**Strengths:**

+ The proposed benchmark dataset is carefully curated to avoid potential data leakage (e.g. proteins similar to test cases are observed during training), and the additional efforts are paid to annotate unlabeled data using UniProt knowledge.
+ It is an interesting research topic to study the influence of function prediction on affinity prediction and vice versa.

**Weaknesses:**

- The comparison experiments do not sufficiently illustrate the effect of each single task on each other task, and thus less insights can be obtained from the current draft.
- Some important baseline models are omitted in performance comparison.
- The proposed protein encoder lacks some novelty, which looks like a simple combination of GearNet and dyMEAN layers.

Detailed in the Question section.

**Questions:**

In general, I am convinced by the benchmark construction process and regard it as a decent contribution in this field. However, in terms of experimental comparison and multi-task learning, I have concerns as below:

1. In current multi-task learning experiments, we can observe obvious performance decay on function prediction tasks after coupling with affinity prediction, which is counter-intuitive. There should be some deep analysis on this phenomenon.

2. Although performance gains are observed on affinity prediction after coupling with function prediction tasks, it is still not clear which specific task (EC, MF, BP or CC) leads to such increase. Authors are suggested to supplement more fine-grained task-coupling results, i.e., LBA&PPI + EC, LBA&PPI + MF, LBA&PPI + BP and LBA&PPI + CC. These results can help us better identify good task combinations.

3. There are some state-of-the-art protein encoders ignored for performance comparison, like CDConv [a] and Full-atom GearNet [b].

4. A more thorough discussion on the technique contributions against GearNet and dyMEAN is suggested.


[a] Fan, Hehe, et al. "Continuous-Discrete Convolution for Geometry-Sequence Modeling in Proteins." ICLR, 2022.

[b] Zhang, Zuobai, et al. "Physics-Inspired Protein Encoder Pre-Training via Siamese Sequence-Structure Diffusion Trajectory Prediction." NeurIPS, 2023.

---

> ### Author Response · Authors · 2023-11-23
> **Official Comment of Authors**
>
> Thank you for the careful review comments and we present the reply as follows:
>
> **Q1: When coupling with affinity prediction tasks, the function prediction performance decays. There should be some deep analysis on this phenomenon.**
>
> A1: We provide a qualitative analysis here. The decay may comes from two aspects: 1) All the samples of single task contains the target property, resulting to less training noise and better performance. While in the multi-task setting, there are many partially labeled samples which does not contain some property. 2) In the multi-task setting, we keep the same parameter size as used in the single-task setting and extend the model's ability to deal with multiple tasks. Actually, to fairly compare their performance, if we compare the performance/task_num of multi-task and single-task models, our multi-task models outperform their single-task counterparts on all of the 6 tasks.
>
> **Q2:Authors are suggested to supplement more fine-grained task-coupling results. (i.e., LBA&PPI + EC, LBA&PPI + MF, LBA&PPI + BP and LBA&PPI + CC)**
>
> A2: Sure, the results are shown below.
>
> |Combination| LBA |  PPI | EC | MF | BP | CC |
> |  :-: |  :-:  | :-:| :-: | :-: | :-: | :-:|
> |   LBA&PPI&EC | 1.758 | 1.064 | 0.628| \- | \- | \- |
> |   LBA&PPI&MF | 1.787 | 1.190 | \- | 0.671| \- |
> |   LBA&PPI&BP| 1.820 | 1.089 | \- | \- | 0.374| \- |
> | LBA&PPI&CC| 1.798 | 1.215 |\- |\- |\- | 0.392|
> | HeMeNet | 1.730 | 1.087 | 0.810 | 0.727 | 0.379 | 0.436|
>
> As shown in the table, different property prediction tasks can improve the performance of LBA and PPI prediction, compared with the single-task settings. However, adding six tasks together will result in an overall better result. We suppose this is because the property prediction tasks shares a high correlation with each other, combining them will benefit our model by increasing the sample size and label diversity.
>
> **Q3: There are some state-of-the-art protein encoders ignored for performance comparison, like CDConv and Full-atom GearNet.**
>
> A3: Thank you for this kind suggestion, and we will compare our model with these baselines in the future.
>
> **Q4: A more thorough discussion on the technique contributions against GearNet and dyMEAN is suggested.**
>
> A4: The key component of our model contains not only the graph encoder but also the task-aware readout method for multiple tasks. **We are the first to solve PPI, LBA, and Property task together in a full-atom way**. The dynamic multi-channel message passing idea comes from dyMEAN, while we include new ralational message passing and separate design for different components of the various input. GearNet only contains single-chain proteins. For our vairous input (single-chain and complex), we design different types of edges to pass heterogenous messages in the network. For the small molecules(optional in the graph), we only construct spatial edges, while for the proteins we include self-loop, sequence and spatial edges. Moreover, the relational message passing of GearNet is **extended to the equivariant setting** in our architecture for the first time. The readout design is also new in the graph encoder. Traditionally, readout methods in GNN are light-weight(i.e.), we proposed a novel task-aware readout method for the first time to give GNN a transformer-based readout function **for the multi-task setting**. **This is totally different from prefix tuning for the transformer encoder because it is used after the struture encoder, only for information aggregation**. Different prompts are designed to notice nodes from chains for property prediction and nodes in the whole graph for affinity prediction.

---

### Official Review · Reviewer_Katf · 2023-10-28

**Soundness:** 2 fair
**Presentation:** 3 good
**Contribution:** 2 fair
**Rating:** 5
**Confidence:** 3

**Summary:**

This paper aims to conduct multi-task learning of protein 3D structures. To do that, the authors build a benchmark named PROMPT integrating the structures and labels from 4 public datasets. The PROMPT dataset contains 3 different types of inputs, including single-chain proteins, protein-protein, and ligand-protein complexes. On top of that, the authors propose a Heterogeneous Multichannel Equivariant Network (HeMeNet) as the base neural architecture of multi-task learning. A task-aware readout mechanism is designed to associate the output head of each task with a learnable task prompt. Extensive experiments compared to a few baselines on the single-task setting and multi-task setting confirm the superiority of HeMeNet.

**Strengths:**

- The authors contribute a new multi-task benchmark by integrating 4 public datasets. Protein data samples are properly processed to obtain fully multi-task labels.
- They propose an Equivariant GNN with structural protein inputs and different readouts for several tasks. Self-attention and learnable task vectors are employed in task-aware readouts.
- Empirical performance on fully labeled test sets demonstrates significantly better performance of the proposal than other baselines.
- The paper is well-written and easy to follow. The experiments are clear.

**Weaknesses:**

- The motivation of this paper can be further enhanced. That is, protein representation learning has been greatly enhanced by a pre-trained mechanism in a self-supervised way. It is not convincing to me why multi-task is still of high research interest. I would like to see more explanation to motivate this paper. Further, some empirical evidence would also be helpful.
- There are previous studies demonstrating similar performance gains for proteins from multi-task sequence models, such as [1][2][3]. I acknowledge that it is non-trivial to extend the success in using sequential information to structural information and build structural benchmarks. Yet the novelty and contribution of building such a benchmark alone, when the proposed HeMeNet is not adequately novel, is not as expected to be published in the main conference of ICLR.
- The HeMeNet network is similar to existing GNN-based protein structure models, like dyMEAN. Also, for task-aware readouts, there are a few recent works inspired by prompt learning and prefix tuning such as [4].
- The experiments do not cover comparisons with prompting LLMs and multi-task protein sequence models. Thus it is not clear how much gain is achieved.

[1] PEER: A Comprehensive and Multi-Task Benchmark for Protein Sequence Understanding

[2] Multi-task learning to leverage partially annotated data for PPI interface prediction

[3] ProteinGLUE multi-task benchmark suite for self-supervised protein modeling

[4] Multi-level Protein Structure Pre-training via Prompt Learning

**Questions:**

Please refer to the limitations.

---

> ### Author Response · Authors · 2023-11-23
> **Official Comment of Authors 1/2**
>
> Thank you for the careful review comments and we present the reply as follows:
>
> **Q1: It is not convincing why multi-task is still of high research interest.**
>
> A1: We propose this multi-task benchmark and model for three reasons:
> 1) First of all, the labeled protein structure for each task is small; simply training on these single tasks may not always provide good results, but combining them can help the model transfer knowledge in different tasks for better performance.
> 2) Meanwhile, the pretain-finetune paradigm has shown its power in the protein representation learning area. However, for each downstream task, the pre-trained model needs to be finetuned separately, which is time and resource consuming.
> 3) More importantly, Our multi-task benchmark does not conflict with the pre-trained models, and researchers can easily fintune a structure-pretrained encoder in a multi-task setting. This setting will bring the model's ability to deal with multiple tasks with only one finetuning procedure. By the way, we can also simply add pre trained sequence-based models' embedding to the structure encoders' input to enhance their performance.
>
> **Q2: What's the significance of the proposed structural dataset when compared to the existing sequence based multitask learning.**
>
> A2: Thanks for mentioning these works, and we will add them to our related works.
> However, for all of the sequence-based benchmarks, their basic unit is amino acids, which can not refine tasks in geometric space, such as interactive tasks in our dataset(i.e., PPI and LBA), and equivariant tasks, such as protein docking. These three multi-task settings are similar to our setting but are different from our idea in many ways.
>
> 1) PEER[1]： Although the tasks are comprehensive, there is no one unified model that handles them all. The multi-task setting is elementary and is one main task + one auxiliary task, which does not require sample matching and additional label merging. This 'multi-task setting' is elementary to train, and the author only reported the performance of the main task. However, we use one model to optimize multiple different tasks simultaneously.
>
> 2) Multitask Learning for PPI interface prediction[2]：The partial label and multi-task settings are similar to ours. However, this dataset only fuses data at the PDB level. Labels with different PDB IDs are neglected, significantly reducing the available training data and making the multi-task learning labels underutilized. In addition, the prediction target was binding surfaces, which do not actually involve affinity prediction for multi-body interactions. In comparison, our benchmark fuses the data in a fine-grained UniProt ID level and collects the affinity labels for LBA and PPI.
>
> 3) ProteinGLUE[3]：This method proposed seven downstream tasks to evaluate the performance of the **pre-trained models**, which are not quite consistent with our scope. These seven tasks are collected separately, and each task requires individual fine-tuning. This pretrain-finetune paradigm does not make the model capable of multi-tasking.
>
> We believe that creating a structured multi-task benchmark is necessary and has many unique advantages. We can not only take this benchmark to train a multi-task model from scratch at the full-atom level but also fine-tune the pre-trained models in a new multi-task way.
>
> [1] PEER: A Comprehensive and Multi-Task Benchmark for Protein Sequence Understanding
>
> [2] Multi-task learning to leverage partially annotated data for PPI interface prediction
>
> [3] ProteinGLUE multi-task benchmark suite for self-supervised protein modeling

---

> ### Author Response · Authors · 2023-11-23
> **Official Comment of Authors 2/2**
>
> **Q3: The model's novelty is not significant due to some similar works, such as dyMEAN and PromptProtein.**
>
> A3: The critical component of our model contains not only the graph encoder but also the task-aware readout method for multiple tasks. **We are the first to solve PPI, LBA, and Property tasks together in a full-atom way**. The dynamic multi-channel message passing idea comes from dyMEAN, while we include new relational message passing and separate designs for different components of the various inputs. We designed different types of edges to pass heterogeneous messages in the network. For the small molecules(optional in the graph), we only construct spatial edges, while for the proteins, we include self-loop, sequence and spatial edges. Moreover, the relational message passing was **extended to the equivariant setting** in our architecture for the first time. Traditionally, readout methods in GNN are lightweight (i.e.); we proposed a novel task-aware readout method for the first time to give GNN a transformer-based readout function **for the multi-task setting**. **This is totally different from prefix tuning for the transformer encoder[4] because it is used after the structure encoder only for information aggregation**. Different prompts are designed to notice nodes from chains for property prediction and nodes in the whole graph for affinity prediction.
>
> **Q4: The experiments do not cover comparisons with prompting LLMs and multi-task protein sequence models. Thus it is not clear how much gain is achieved.**
>
> A4: As stated in [5], LLMs such as GPT-4 does not perform well when calculating the binding affinity in drug discovery. To demonstrate this, we conducted experiments of prompting the advanced GPT-4-turbo-1106 version. To get a higher performance, we follow the paper[5] to design prompts for affinity prediction. The experimental settings are as follows:
>
> ```
> System Prompt: You are a drug assistant and should be able to help with drug discovery tasks. Given the SMILES sequence of a drug and the FASTA sequence of a protein target, you need to calculate the binding affinity score. You can think step-by-step to get the answer and call any function you want. You should try your best to estimate the affinity with tools. The output should be a float number, which is the estimated affinity score without other words.
>
> Prompt:
> Example c1:
> CC[C@H](C)[C@H](NC(=O)OC)C(=O)N1CCC[C@H]1c1ncc(-c2ccc3cc(-c4ccc5[nH]c([C@@H]6CCCN6C(=O)[C@@H](NC(=O)OC)[C@@H](C)OC)nc5c4)ccc3c2)[nH]1,
> MDSIQAEEWYFGKITRRESERLLLNAENPRGTFLVRESETTKGAYCLSVSDFDNAKGLNVKHYKIRKLDSGGFYITSRTQFNSLQQLVAYYSKHADGLCHRLTTVCP
> 11.52
>
> Example 2:
> [H]C1:C([H]):C(S(=O)(=O)N([H])[H]):C([H]):C([H]):C:1/N=N/N1C([H])([H])C([H])([H])C([H])([H])C([H])([H])C([H])([H])C1([H])[H],
> HWGYGKHNGPEHWHKDFPIAKGERQSPVDIDTHTAKYDPSLKPLSVSYDQATSLRILNNGHAFNVEFDDSQDKAVLKGGPLDGTYRLIQFHFHWGSLDGQGSEHTVDKKKYAAELHLVHWNTKYGDFGKAVQQPDGLAVLGIFLKVGSAKPGLQKVVDVLDSIKTKGKSADFTNFDPRGLLPESLDYWTYPGSLTTPPLLECVTWIVLKEPISVSSEQVLKFRKLNFNGEGEPEELMVDNWRPAQPLKNRQIKASFK
> 6.5
>
> Example 3:
> [H]/C1=C(\\C([H])([H])C(=O)N([H])C2:C([H]):C([H]):C(S(=O)(=O)N([H])[H]):C([H]):C:2[H])C2:C([H]):C([H]):C(OC([H])([H])[H]):C([H]):C:2OC1=O,
> HWGYGKHNGPEHWHKDFPIAKGERQSPVDIDTHTAKYDPSLKPLSVSYDQATSLRILNNGHAFNVEFDDSQDKAVLKGGPLDGTYRLIQFHFHWGSLDGQGSEHTVDKKKYAAELHLVHWNTKYGDFGKAVQQPDGLAVLGIFLKVGSAKPGLQKVVDVLDSIKTKGKSADFTNFDPRGLLPESLDYWTYPGSLTTPPLLECVTWIVLKEPISVSSEQVLKFRKLNFNGEGEPEELMVDNWRPAQPLKNRQIKASFK
> 8.05
>
> for smiles, fasta in messages
>     Test input:
>     {smiles},
>     {fasta}
> ```
> The test result is as follows (measured with RMSE):
> |Model | LBA |  PPI |
> |  :-: |  :-:  | :-:|
> |   HeMeNet | 1.730 | 1.087 |
> |   DyMEAN  | 1.777 | 1.725 |
> |   GearNet | 2.193 | 1.275 |
> | GPT-4-turbo | 2.347 | 1.654 |
>
> As shown in the results, our model significantly outperforms GPT-4 in both LBA and PPI prediction, showing the advantage of incorporating structural information.
>
> [4] Multi-level Protein Structure Pre-training via Prompt Learning
>
> [5] The Impact of Large Language Models on Scientific Discovery: a Preliminary Study using GPT-4.

---

### Official Review · Reviewer_STD6 · 2023-10-30

**Soundness:** 3 good
**Presentation:** 2 fair
**Contribution:** 1 poor
**Rating:** 3
**Confidence:** 3

**Summary:**

The paper introduces a benchmark dataset for atom-level multi-task learning in the context of proteins. Accompanying this, the authors present a heterogeneous equivariant graph neural network designed to support multi-task learning. The newly introduced dataset, named Protein Multiple Task (PROMPT), comprises 31,887 samples and encompasses six distinct tasks, spanning ligand binding affinity predictions, protein-protein interactions, enzyme commission number predictions, as well as molecular function, biological process, and cellular component properties.

The proposed neural network, referred to as the Heterogeneous Multichannel Equivariant Network (HeMeNet), is adept at learning through heterogeneous message passing across various edge types while maintaining invariance to E3 transformations. HeMeNet incorporates an attention-based task-aware readout mechanism, enabling simultaneous learning of different tasks. The architecture is optimized through the summation of losses from the multiple tasks it addresses.

**Strengths:**

1. The benchmark dataset can be convenient and practical for studying protein properties.

2. The experiment results of HeMeNet are strong, as presented in Table1.

3. The paper is easy to follow.

**Weaknesses:**

1. The proposed benchmark, in my evaluation, does not exhibit novelty in terms of introducing new protein structures or bringing forth innovative tasks for protein learning. PROMPT primarily amalgamates existing protein datasets and employs a matching pipeline to transfer labels between enzyme commission and properties of gene products.\

2. The proposed network, HeMeNet, lacks novelty. The concept of heterogeneous message passing, as presented, is simply realized by incorporating additional edge representations into the messages. This core idea appears to be adapted from the existing dyMEAN approach.

3. The computation of the training loss as the simple summation of all sub-losses for multi-task learning is overly simplistic.

**Questions:**

1. I would like to propose an experiment to investigate the internal transfer of information within the tasks. For instance, in your experiments involving ligand binding affinity (LBA) and protein-protein interactions (PPI) predictions, you set the weight of PPI to zero during training, effectively isolating the loss of LBA. I suggest conducting an experiment where the predictions of PPI are optimized on the test and validation data, even with a zero weight during training. This would allow us to explore the extent to which information from one task can be transferred and benefit the prediction of another task during testing and validation.

---

> ### Author Response · Authors · 2023-11-23
> **Official Comment by Authors**
>
> Thank you for the careful review comments and we present the reply as follows:
>
> **Q1: The benchmark primarily amalgamates existing protein datasets and employs a matching pipeline to transtfer labels between enzyme commission and properties of gene products.**
>
> A1: The main motivation of our dataset is to help structure related tasks to benefit each other. Actually, it is not straightforward to combine different structure-based datasets together for two reasons: 1) Different datasets contain different levels of samples, including single-chain samples(tertiary structure) and complex samples(quarternary structure). Directly combining these data without a unified matching method is meaningless. 2)**The former tasks, such as PPI and LBA, only focus on the interface and pocket, hindering the ability to combine each other**. We first extend these tasks in a whole protein and full-atom setting, then combine them on this unified level. 3) The labels of these samples are also different. The property tasks(EC, BP, MF, CC) label single chains, while the affinity tasks(PPI, LBA) label complexes. For a complex containing multiple chains, we want to annotate the property for as many chains as possible. Therefore, a simple matching based on the PDB ID (The standard name for samples in the six tasks) does not work for our setting; we need the Uniprot ID as the bridge to cross annotate each chain. Creating such a partially labeled structure dataset by our labeling method is the first attempt at structure-based protein representation learning. We believe that creating a structure multitask benchmark is necessary with many unique advantages. We can not only take this benchmark to train a multitask model from scratch at the full-atom level, but also finetune the pretrained structure encoders in a new multi-task way.
>
> **Q2: The proposed HeMeNet lacks novelty.**
>
> A2: While the dynamic multi-channel message passing idea comes from dyMEAN, we include new relational message passing and separate designs for different components of the various inputs. We designed different types of edges to pass heterogeneous messages in the network. We only construct spatial edges for the small molecules(optional in the graph), while we include self-loop, sequence and spatial edges for the proteins. Moreover, the relational message passing was extended to the equivariant setting in our architecture for the first time.
>
> Besides the graph encoder (HeMeNet), the task-aware readout method for multiple tasks is also a critical component of our model, which for the first time gives GNN a transformer-based readout function for the multi-task setting. Different prompts are designed to notice nodes from chains for property prediction and nodes in the whole graph for affinity prediction. As a result, we are the first to solve PPI, LBA, and property tasks together in a full-atom way.
>
> **Q3: The computation of the training loss is too simple.**
>
> A3: Classic GNN methods use a sum/mean readout method for a simple task. In our work, we developed a novel task-aware readout method that contains task prompts and attention mechanisms to coordinate different tasks automatically. After this readout function, adding multiple losses with different weights can provide a strong enough performance in the setting. The ablation study for the readout strategy shows the effectiveness of the proposed task-aware readout, showcasing the correctness of our design for multi-task learning. The results show that the training loss fits the current task and setting, and in multi-task learning, there are also some recent works that use this simple loss combination[1].
>
> **Q4: We can add an experiment to investigate the internal transfer of information within the tasks.**
>
> A4: Sure, we add the experiment as suggested together with more selected ratios. For each run, we keep the samples from the other 5 tasks and add different ratios of PPI samples, and the results (RMSE) are as follows:
> | Ratio|  PPI  |
> |  :-: |  :-:  |
> |   0  | 7.764 |
> | 1/12 | 1.651 |
> | 1/6  | 1.161 |
> | 1/3  | 1.126 |
> | 1/1  | 1.087 |
>
> As we can see from the table above, completely removing the PPI data in the training set will cause significant damage to the performance. We infer this phenomenon is **because the ppi task prompt(which is randomly initialized) is not tuned at all** and the task-specific mlp is also not tuned. However, we only need 1/12 of the training set PPI to get a reasonable performance demonstrating the knowledge transfer from other tasks. As the ratio of the PPI training sample increases, the PPI performance monotonically increases.
>
> [1] Multi-level Protein Structure Pre-training via Prompt Learning

---

### Official Review · Reviewer_y1Y9 · 2023-11-01

**Soundness:** 2 fair
**Presentation:** 3 good
**Contribution:** 2 fair
**Rating:** 3
**Confidence:** 3

**Summary:**

This paper presents a multitask learning framework for protein property prediction. Specifically, a dataset is constructed consisting 6 tasks from 4 sources and an equivariant Transformer is proposed to extract task-specific embeddings in a multitask setting. Experimental results demonstrate the effectiveness of the proposed strategy.

**Strengths:**

+ The formulation is clearly motivated and could benefit a wide range of biochemistry problems
+ The idea is easy to understand and the paper is well written
+ Protein representations seems to benefit from the proposed multitask learning paradigm

**Weaknesses:**

- The current presentation of the datasets and their construction appears to be straightforward. It is essential for the readers to understand the nature and significance of the studied properties: EC, MF, BP, and CC. It would be helpful if the authors could elucidate whether these properties are mere one-hot encodings of certain database indices or if they carry deeper biological or computational significance.
- It would be constructive if the authors provide insight into how these properties (EC, MF, BP, CC) interact or interplay with each other. Explaining this can give a clearer understanding of how collectively these properties can enhance protein representation learning.
- The claim that the Transformer is E(3)-equivariant is not evident from the text. A detailed proof or a more comprehensive explanation of this feature is required.
- It would be beneficial for the readers to understand the motivation and design principles behind the geometric relation extractor and message scaler. Providing the rationale can establish the unique contribution of this work and its difference from existing architectures.
- The experimental results seem to be marginal on some tasks. The multitask baselines employed all rely on sum pooling for aggregating task-specific embeddings. A stronger baseline or justification for the chosen approach would give more credibility to the reported results.

**Questions:**

Apart from points in the "weakness" part, the authors may need to provide a table summarizing the dataset with more statistics, e.g., the ratio of missing targets. Also, there are some missing details in the model architecture, e.g., $\phi_m, \phi_x, \epsilon$.

---

> ### Author Response · Authors · 2023-11-23
> **Official Comment by Authors 1/2**
>
> **Q1: The construction of dataset and the biological significance.**
>
> A1: It is not straightforward to combine different structure-based datasets together for three reasons: 1) Different datasets contain different levels of samples, including single-chain samples(tertiary structure) and complex samples(quarternary structure). Directly combining these data without a unified matching method is meaningless. 2)The former tasks, such as PPI and LBA, only focus on the interface and pocket, hindering the ability to combine each other. We first extend these tasks in a whole protein and full-atom setting, then combine them on this unified level. 3) The labels of these samples are also different. The property tasks(EC, BP, MF, CC) labels single chains, while the affinity tasks(PPI, LBA) label complexes. For a complex containing multiple chains, we want to annotate the property for as many chains as possible. Therefore, a simple matching based on the PDB ID (The standard name for samples in the six tasks) does not work for our setting; we need the Uniprot ID as the bridge to cross and annotate each chain. We want to clarify that the labels for properties are not mere one-hot encodings of the indices while they contain biological meaning. For the relationship between different tasks, here is an intuitive explanation: The selected protein property tasks describe the functions of a protein when interacting with other targets(molecules or proteins). For example, the EC number describes which type of enzyme the protein belongs to in the biochemical reactions, which imply the protein's binding affinity with some type of target. Based on this observation, we assume that multi-task training on these tasks can improve protein representation learning quality and further benefit the affinity prediction tasks. The strong results showcase our assumptions.
>
> **Q2: How these properties (EC, MF, BP, CC) interact or interplay with each other?**
>
> A2: In Figure 3 of our draft, we showcase the correlation of different tasks' prompts. According to the results, we can conclude that the prompts for tasks of the same level (complex level affinity and single-chain level property) have high intra-level correlation while the prompts have low inter-level correlation, which indicates that our model performs harmonious multi-task training. From the biological perspective, the four tasks describe different properties of the same protein in different directions. EC number classifies the protein into different Enzyme families, each of which can catalyze different reactions. The MF task describes the function of an individual gene product, such as carbohydrate-binding or ATP hydrolase activity. The BP task annotates the ordered combination of molecular functions to achieve a broader biological function, such as mitosis or purine metabolism. The CC task annotates subcellular structure, location, and macromolecular complexes such as nucleolus, telomeres, and complexes that recognize initiation. These four tasks describe the protein's function(including binding) from different perspectives, and combining them can help us enhance protein representation learning.
>
> **Q3: The E(3) equivariance of the transformer is not proved.**
>
> A3: The HeMeNet encoder of our model is E(3) equivariant, and the TAR module is E(3) invariant. We enhance the representation ability of dyMEAN by adding relational message passing and relational coordinate update, keeping its original equivariance. Therefore, the proof of the E(3) equivariance of HeMeNet is similar to that of dyMEAN. The encoder outputs both invariant node features and equivariant coordinates for different downstream tasks. Actually, we don't introduce E(3) equivariance tasks in this work, so our task-aware readout transformer is E(3) invariant with the invariant node feature as its input. Incorporating new E(3) equivariance tasks as well as the corresponding E(3) Task-aware readout functions can be the possible future work.
>
> **Q4: Explain further about the geometric ralation extractor and message scalar.**
>
> A4: As shown in the paper's Figure 2, our model is proposed to deal with various inputs and complete six different downstream tasks. In our dataset, a complex can contain up to 15000 atoms, if we pass messages among atoms, the computational complexity will be very high. Therefore, we need a way to pass messages efficiently while keeping the full-atom coordinate information. Some of the input contains a ligand while others do not, and some tasks are chain-level while others are complex-level. To deal with this heterogeneous input and output situation, we first conduct the geometric relation extractor to pass messages among atoms in each nodes(for protein, they are amino acids and for ligand, they are atoms). And a geometric message scaler is used to dynamicly pass messages between amino acids that contains different atoms.

---

> ### Author Response · Authors · 2023-11-23
> **Official Comment by Authors 2/2**
>
> **Q5: More information about the dataset and the model architecture.**
>
> A5: Below, we represent a more detailed version of our dataset as an extension of our table shown in the paper's Appendix A.
>
> |Task Name | #Train samples |  #Validataion samples| # Test samples | Train label missing ratio | Average seq length | Average atoms |Chains |
> |  :-: |  :-:  | :-:| :-: | :-: | :-: | :-:| :-: |
> |  Multi-task | 30904 | 516 | 467| 99.9% | 325.67 | 2537.68 | Mixed |
> |   LBA | 3247 | 493 | 452 | 89.5% | 425.29 | 3337.21 |Multi |
> |   PPI | 1976 | 23 | 15 | 93.6% |846.53 |6604.35 | Multi |
> | EC| 15025 | 516 | 467 | 51.4% | 331.68|2585.91 | Single |
> | MF | 22997 | 516 | 467 | 25.6% | 311.18|2420.01| Single |
> | BP | 21626 | 516 | 467 |  30.0%| 305.90|2382.08| Single|
> | CC | 10543 | 516 | 467 | 65.9% | 302.47| 2346.91|Single|
>
> As shown in this table, our dataset contains graphs with various sizes and a multi-task training set with 99% partially labeled data, making the samples noisy and the task hard to solve. With the careful design of our network and task-aware readout, we can develop a model that benefits from this dataset.
>
> For the model's detailed information, as stated in the paper's Section 4.2, $\phi_m$ and $\phi_x$ are multi-layer perceptros with one hidden layer. And $\epsilon$ is a parameter to avoid a denominator of zero, set to 1e-3.
>
> **Q6: Further justification of the experiment baselines.**
>
> A6: We divide the structure-based baselines into two types: invariant models(i.e., SchNet, GearNet) and equivariant models (i.e., EGNN, GVP, dyMEAN). We do not choose sequence-based models because this is a structural multi-tak benchmark, and we want to compare different structure encoders. As there are no structure-based multi-task models, we choose some SOTA models in different subtasks for comparison. GearNet is a SOTA pretraining structure encoder in protein property prediction tasks, which is on par with the SOTA sequence-based models. Comparing these invariant and equivariant models in both single-task and multi-task demonstrates our model's performance.

---

### Meta-Review · Area_Chair_9Stm · 2023-12-07

**Metareview:**

The paper proposes a framework and benchmark for structured-based protein function prediction in the multitask setting. This is a pertinent topic and the benchmark presented in the paper is certainly valuable.

The AC carefully examined the author feedback to the reviews and while it clarifies several points, there are some important concerns remaining. In particular, as all reviewers indicated, the proposed approach is of limited novelty. It is therefore especially critical to conduct a thorough comparison. In particular, as suggested by reviewer N1C4.  it would be important to compare against state-of-the-art protein encoders such as CDConv and Full-atom GearNet.

**Justification For Why Not Higher Score:**

The AC and reviewers all find that the contributions are of limited novelty.  Comparison against important baselines are missing as the authors themselves acknowledged.

**Justification For Why Not Lower Score:**

N/A

---

### Decision · Program_Chairs · 2024-01-16

Reject